# Genetic Aspects of Mammographic Density Measures Associated with Breast Cancer Risk

**DOI:** 10.3390/cancers14112767

**Published:** 2022-06-02

**Authors:** Shuai Li, Tuong L. Nguyen, Tu Nguyen-Dumont, James G. Dowty, Gillian S. Dite, Zhoufeng Ye, Ho N. Trinh, Christopher F. Evans, Maxine Tan, Joohon Sung, Mark A. Jenkins, Graham G. Giles, John L. Hopper, Melissa C. Southey

**Affiliations:** 1Centre for Epidemiology and Biostatistics, Melbourne School of Population and Global Health, The University of Melbourne, Parkville, VIC 3051, Australia; shuai.li@unimelb.edu.au (S.L.); nguk@unimelb.edu.au (T.L.N.); jdowty@unimelb.edu.au (J.G.D.); gillian.dite@gtglabs.com (G.S.D.); zhoufengy@student.unimelb.edu.au (Z.Y.); nhut.trinh@unimelb.edu.au (H.N.T.); cfevans@unimelb.edu.au (C.F.E.); m.jenkins@unimelb.edu.au (M.A.J.); graham.giles@cancervic.org.au (G.G.G.); 2Centre for Cancer Genetic Epidemiology, Department of Public Health and Primary Care, University of Cambridge, Cambridge CB1 8RN, UK; 3Precision Medicine, School of Clinical Sciences at Monash Health, Monash University, Clayton, VIC 3168, Australia; tu.nguyen-dumont@monash.edu (T.N.-D.); melissa.southey@monash.edu (M.C.S.); 4Department of Clinical Pathology, The University of Melbourne, Parkville, VIC 3051, Australia; 5Genetic Technologies Limited, Fitzroy, VIC 3065, Australia; 6Electrical and Computer Systems Engineering Discipline, School of Engineering, Monash University Malaysia, Bandar Sunway 47500, Malaysia; maxine.tan@monash.edu; 7School of Electrical and Computer Engineering, The University of Oklahoma, Norman, OK 73019, USA; 8Department of Public Health Sciences, Division of Genome and Health Big Data, Graduate School of Public Health, Seoul National University, Seoul 08826, Korea; jsung@snu.ac.kr; 9Cancer Epidemiology Division, Cancer Council Victoria, Melbourne, VIC 3004, Australia

**Keywords:** Altocumulus, breast cancer, Cirrocumulus, Cumulus, genome-wide association studies, mammographic density, mammogram risk score, single nucleotide polymorphism

## Abstract

**Simple Summary:**

Mammogram risk scores, based on the area of a mammogram covered by white or bright areas defined at different levels of pixel brightness, predict breast cancer risk. They are correlated in relatives, and this might be due to genetic factors that cause breast cancer. We found that the known genetic markers associated with breast cancer risk were weakly correlated with the studied mammogram risk scores. Our findings suggest that less than 1% of the variance of the studied mammogram risk scores is explained by the known genetic markers associated with breast cancer risk. Discovering the genetic determinants of the bright, not white, regions of a mammogram could reveal new information about the genetic causes of breast cancer.

**Abstract:**

Cumulus, Altocumulus, and Cirrocumulus are measures of mammographic density defined at increasing pixel brightness thresholds, which, when converted to mammogram risk scores (MRSs), predict breast cancer risk. Twin and family studies suggest substantial variance in the MRSs could be explained by genetic factors. For 2559 women aged 30 to 80 years (mean 54 years), we measured the MRSs from digitized film mammograms and estimated the associations of the MRSs with a 313-SNP breast cancer polygenic risk score (PRS) and 202 individual SNPs associated with breast cancer risk. The PRS was weakly positively correlated (correlation coefficients ranged 0.05–0.08; all *p* < 0.04) with all the MRSs except the Cumulus-white MRS based on the “white but not bright area” (correlation coefficient = 0.04; *p* = 0.06). After adjusting for its association with the Altocumulus MRS, the PRS was not associated with the Cumulus MRS. There were MRS associations (Bonferroni-adjusted *p* < 0.04) with one SNP in the *ATXN1* gene and nominally with some *ESR1* SNPs. Less than 1% of the variance of the MRSs is explained by the genetic markers currently known to be associated with breast cancer risk. Discovering the genetic determinants of the bright, not white, regions of the mammogram could reveal substantial new genetic causes of breast cancer.

## 1. Introduction

Conventionally, mammographic density refers to the white or bright regions on a mammographic image and can be measured using the computer software CUMULUS [1]. In this study, we name mammographic density based on this definition as the Cumulus density measure and the percentage dense area, defined as the dense area divided by the total breast area and expressed as a percentage, as the Cumulus-percent density measure.

Over the last few years, we have introduced other mammographic density measures based on defining density at, in effect, higher pixel brightness thresholds [2,3,4,5]. In this study, we name the Altocumulus density measure as the dense area defined by the bright regions and the Cirrocumulus density measure as the dense area defined by the brightest regions.

Age and body mass index (BMI) are confounders of the association between mammographic density measures and breast cancer risk, and these risk factors are always adjusted for. Therefore, when we are talking about a mammographic density measure in terms of its association with breast cancer risk, we are essentially talking about the mammographic density measure adjusted for age and BMI.

We define a mammogram risk score (MRS) to be the residual of a mammographic measure (transformed to approximate normality if needed) after making an adjustment for age and BMI. In this study, we name the MRSs calculated from Cumulus, Cumulus-percent, Altocumulus, and Cirrocumulus density measures as the Cumulus MRS, Cumulus-percent MRS, Altocumulus MRS, and Cirrocumulus MRS, respectively.

The Cumulus MRS and Cumulus-percent MRS have been established as predictors of breast cancer risk [6]. From a series of studies of Australian and Korean and now North American women, we have found consistently that the Altocumulus MRS and Cirrocumulus MRS provide more information on breast cancer risk than does the Cumulus MRS [2,3,4,5,7,8]. Furthermore, the risk association with the Cumulus MRS attenuates and often becomes null after adjusting for the Altocumulus MRS or the Cirrocumulus MRS [8,9].

The potential role of familial, including genetic, factors in explaining the variance of the MRSs has been considered for more than two decades. Studies have found that the familial correlations in both the Cumulus MRS and Cumulus-percent MRS were about 0.6–0.7 for monozygotic (MZ) twin pairs and about one-half of this (~0.3) for dizygotic (DZ) twin pairs and sister pairs [10,11,12,13]. Under the equal environments assumption of the classic twin model, additive genetic factors explain about 60–70% of the variance of the Cumulus MRS and Cumulus-percent MRS [10,11,12,13].

We recently found that the familial correlation in the Altocumulus MRS was about 0.6 for MZ twin pairs and about 0.25 for DZ twin pairs and sister pairs, while the familial correlation in the Cirrocumulus MRS was about 0.4 for MZ twin pairs and about 0.2 for DZ twin pairs and sister pairs. This is consistent with additive genetic factors explaining about 60% of the variance of the Altocumulus MRS and 40% of the variance of the Cirrocumulus MRS [13].

Based on the relationship between familial risk factors and familial aggregation of diseases [14,15,16,17], we predicted that the four MRSs individually explain, in a statistical sense, around 5–10% of the overall familial risk of breast cancer, which is about one-quarter to one-half as much of the familial risk of breast cancer that is explained by the latest polygenic risk score (PRS) based on 313 single-nucleotide polymorphisms (SNPs) [18].

Studies trying to find the genetic determinants of MRSs have been only conducted for the Cumulus MRS and Cumulus-percent MRS. To date, genome-wide association studies (GWAS) have found SNPs at 33 loci to be associated with the Cumulus MRS and SNPs at 25 loci with the Cumulus-percent MRS [19,20,21,22,23,24,25]. However, these SNPs only explain about 2–4% of the variance of the Cumulus MRS and Cumulus-percent MRS [19,25]. About 10% of the SNPs associated with breast cancer have also been found to be nominally associated with the Cumulus MRS and Cumulus-percent MRS [26].

In this study, we aimed to investigate the association between the MRSs and the polygenic determinants of breast cancer risk. We investigated the latest breast cancer PRS and the individual SNPs that had been found to be associated with breast cancer risk from GWAS.

## 2. Materials and Methods

### 2.1. Study Sample

We used data from the Australian Mammographic Density Twins and Sisters Study (AMDTSS) [12,13]. Briefly, female twin pairs aged 40–70 years and without a prior diagnosis of breast cancer were recruited through the Australian Twin Registry. Participating twins completed a questionnaire and gave permission to access their mammograms. They were also asked to seek permission from any eligible sisters to be invited to participate in the study. Participants completed telephone-administered questionnaires, which collected demographic information and self-reported weight, height, smoking history, alcohol consumption, reproductive history, use of oral contraceptives, use of hormone replacement therapy, and personal and family history of cancer. Participants also donated a blood sample. Participants who had mammographic density measurements and GWAS data were included in this analysis. The sample included 2559 twins and sisters from 1159 families, comprising 584 MZ twin pairs, 318 DZ twin pairs, and 755 non-twin sisters. For twin pairs, their zygosity was determined using genotype data. All participants gave written informed consent, and the study was approved by the Human Research Ethics Committee of the University of Melbourne.

### 2.2. Mammographic Density Measurements

All available episodes of mammograms were retrieved with the participants’ written consent, mostly from Australian BreastScreen services but also from private clinics and private hospitals. We also retrieved mammograms from the participants themselves. The craniocaudal views for left and right breasts were selected and digitized using the Lumysis 85 scanner at the Australian Mammographic Density Research Facility. For each woman, the most recent right breast craniocaudal view was selected for mammographic density measurement, and the left breast craniocaudal view was selected if the right breast mammogram was missing or unavailable. Mammographic measurements were performed using CUMULUS 4.0 (University of Toronto, Toronto, ON, Canada), a computer-assisted thresholding technique, after randomization and blind to information, by three independent measurers (T.L.N., H.N.T., C.F.E.) with high repeatability; see, for example, [27]. Total breast area and dense areas, defined by the Cumulus, Altocumulus, and Cirrocumulus density measures, respectively, were measured. The Cumulus-percent density measure was calculated as the Cumulus density measure divided by the total breast area. As in our previous studies [8,13], we also subtracted the Altocumulus density measure from the Cumulus density measure to derive a measure that refers to the white, not bright, regions on a mammogram and called this measure the Cumulus-white density measure.

### 2.3. Genome-Wide SNP Genotyping, Quality Control, and Imputation

Germline DNA extracted from the blood samples was used to perform genome-wide SNP genotyping using the OncoArray (Illumina) [28] and following the standard measurement and genotyping-calling procedures.

For duplicated samples, we excluded the ones with a lower call rate. We excluded samples with a call rate < 95%, extreme heterozygosity (4.89 standard deviations from the mean), or sex discrepancy. Ancestry was computed using a principal component analysis, and women of non-European ancestry were excluded by visualizing the multidimensional scaling plot of the first two principal components, with reference to the 1000 Genomes Project populations (629 individuals, August 2010 release). We excluded SNPs with a call rate < 95%, not in Hardy–Weinberg Equilibrium (*p* < 10^−7^), with a minor allele frequency (MAF) <0.01, or on X chromosome. A total of 399,364 SNPs passed the quality control.

Imputation was conducted using Minmac4 on the Michigan Imputation Server (https://imputationserver.sph.umich.edu/index.html (accessed on 14 September 2021)) with the 1000 Genomes Project Phase 3 (Version 5) as the reference panel. Overall, 14.0 million SNPs with a MAF > 0.1% and an imputation quality score > 0.3 were imputed.

### 2.4. Breast Cancer PRS

All the 313 SNPs of the breast cancer PRS [18] were included in the imputed dataset, and they were extracted from the dataset to calculate the PRS. In our dataset, the 313 SNPs had a minimum imputation quality score of 0.67 and a minimum MAF of 0.2%. The PRS was calculated as
(1)$$PRS=\overset{313}{\underset{i=1}{\sum }}\beta_{i}X_{i}$$
where *β_i_* (*i* = 1,…, 313) is the weight of the *i*th SNP, and *X_i_* is the number of effect alleles of the *i*th SNP. We calculated the PRS for overall breast cancer, estrogen receptor negative (ER−) breast cancer, and estrogen receptor positive (ER+) breast cancer, respectively, using the relevant weights for each PRS reported by Mavaddat et al. [18].

### 2.5. Statistical Methods

The Box–Cox procedure was used to test the normality of the distributions of the mammographic density measures and if necessary to select an appropriate power transformation. As a result, the Cumulus, Cumulus-percent, Altocumulus, Cirrocumulus, and Cumulus-white density measures were cube root, square root, cube root, log, and log transformed, respectively. The five density measures were then used to calculate their relevant MRSs, which were the standardized residuals in the transformed density measures after adjusting for age and the inverse of BMI.

We estimated the Pearson correlations between the MRSs and the PRSs. A total of 15 correlations (5 MRSs × 3 PRSs) were estimated.

We used a mixed-effect model to estimate the associations (regression coefficients) between the MRSs and the PRSs, in which an MRS was the outcome variable, and a PRS was the predictor variable. The first 10 principal components of the GWAS data were adjusted in the model as fixed effects, and family and zygosity were modeled as random effects to account for the relatedness between the study subjects. Given age and BMI were already adjusted when calculating the MRS, and they were not related to the PRSs, they were not included in the regression analyses. A total of 15 associations (regression coefficients) were estimated. For each MRS, we also estimated its association with the PRSs after adjusting for other MRSs, using the same mixed-effect model with other MRSs as additional fixed effects. The regression coefficients were reported using the odds ratio per adjusted standard deviation (OPERA) method [29], i.e., as the change in an MRS per standard deviation change in the residuals of the PRS after it had been adjusted for all covariates.

We used the Bayesian Information Criterion (BIC) to compare the evidence strength between the MRSs for their correlations/associations with the PRSs.

We estimated the association between the MRSs and the SNPs known to be associated with breast cancer risk, using the same mixed-effect model with the PRS being replaced as the SNP. Kappor et al. [30] provided a list of 205 SNPs considered to be associated with breast cancer risk either by GWAS or fine mapping of associated regions. Of these, 202 SNPs were available in our dataset and included in the analysis.

All statistical analyses were conducted using R 4.0.2 (R Core Team, Vienna, Austria).

## 3. Results

### 3.1. Sample Characteristics

Table 1 shows the characteristics, the PRSs, and mammogram density measures for the study sample. The study sample had an age range of 30 to 80 years and an average age of 54.0 years (standard deviation: 8.4 years) and an average BMI of 26.1 kg/m^2^ (standard deviation: 5.2 kg/m^2^). MZ twins, DZ twins, and non-twins did not differ substantially from one another in age, BMI, or any of the PRSs or mammogram density measures.

### 3.2. Correlations between the MRSs and PRSs

Table 2 shows that the MRSs and PRSs were positively and weakly correlated. The correlation coefficients were ~0.06 (standard error: ~0.02) for all combinations of the MRSs and the PRSs, and all the correlation coefficients were statistically significant (*p* < 0.05) except those of the Cumulus-white MRS with the PRS for overall breast cancer or with the PRS for ER+ breast cancer. For each MRS, there was no substantial difference between the correlation coefficients with the three PRSs. For each PRS, the correlation coefficient point estimates were the largest for the Altocumulus MRS, less for the Cumulus-percent MRS and Cumulus MRS, and the smallest for the Cirrocumulus MRS and Cumulus-white MRS, although their 95% confidence intervals (CIs) overlapped with each other. These results suggest that the PRSs explained less than 1% of the variance of the MRSs.

For each PRS, the BIC for the correlation with the Altocumulus MRS was smaller than the BIC for the correlation with the Cumulus-white MRS. Note that, for both the PRS for overall breast cancer and PRS for ER+ breast cancer, the smallest BIC was observed for correlations with the Altocumulus MRS, and the largest BIC was observed for the correlations with the Cumulus-white MRS. The BIC differences between the correlations with the Altocumulus MRS and Cumulus-white MRS ranged from 6.78 to 15.54 across the three PRSs, suggesting that there was strong evidence that the correlations of the PRSs with the Altocumulus MRS were stronger than those with the Cumulus-white MRS.

### 3.3. Associations between the MRSs and PRSs

Table 3 shows the associations between the MRSs and PRSs from the mixed-effect linear model, which adjusted for the first 10 principal components based on genome-wide SNP data and took into account the familial relatedness between the study samples. The MRSs and PRSs were positively associated. All the associations were nominally significant (*p* < 0.05) except those for the Cumulus-white MRS with the PRS for overall breast cancer and with the PRS for ER+ breast cancer. For the Cumulus MRS, the association point estimates were similar across the three PRSs, while for both the Cumulus-percent MRS and Cumulus-white MRS, the association point estimate with the PRS for ER− breast cancer was the largest. For both the Altocumulus MRS and Cirrocumulus MRS, the association point estimates were similar between the PRS for overall breast cancer and the PRS for ER+ breast cancer, and the smallest for the PRS for ER− breast cancer.

For each PRS, the smallest BIC was observed for the association with the Altocumulus MRS. The BIC differences between the associations with the Altocumulus MRS and Cumulus-white MRS ranged from 7.43 to 16.97 across the three PRSs, suggesting that there was strong evidence that the associations of the PRSs with the Altocumulus MRS were stronger than those with the Cumulus-white MRS.

Table 4 shows that, overall, for the Cumulus MRS, Cumulus-percent MRS, Cirrocumulus MRS, and Cumulus-white MRS, their associations with the breast cancer PRSs became null after adjusting for the other MRSs. Only for the Altocumulus MRS did it remain associated after adjusting for the other MRSs (Cumulus MRS, Cirrocumulus MRS, or Cumulus MRS and cirrocumulus MRS together). Compared with the associations between the Altocumulus MRS and the breast cancer PRSs in Table 3, after adjusting for the other MRSs, the associations with the PRS for overall breast cancer decreased by 57% to 75%, those with the PRS for ER− breast cancer decreased by 51% to 80%, and those with the PRS for ER+ breast cancer decreased by 58% to 73%.

### 3.4. Associations between the MRSs and SNPs Known to Be Associated with Breast Cancer Risk

Appendix A shows the associations between the MRSs and the 202 SNPs known to be associated with breast cancer risk. After Bonferroni adjustment for 202 tests, only one SNP, rs3819405 in the *ATXN1* locus, was associated with Cumulus MRS, Altocumulus MRS, and Cumulus-white MRS, respectively (all nominal *p* < 1.7 × 10^−4^), and no SNP was found to be associated with the Cumulus-percent MRS or Cirrocumulus MRS. The numbers of SNPs with a nominal *p* < 0.05 were 36, 26, 25, 18, and 34 for the Cumulus MRS, Cumulus-percent MRS, Altocumulus MRS, Cirrocumulus MRS, and Cumulus-white MRS, respectively, all more than the number expected by chance alone, i.e., 202 × 0.05 = 10.1 (all *p* < 0.02). This suggests that there was some association between one or more of the breast cancer SNPs and the MRSs. However, only two-thirds of the SNP associations were in the same direction as the direction of the association between relevant MRS and breast cancer risk: 24, 18, 17, 10, and 22 SNPs for the Cumulus MRS, Cumulus-percent MRS, Altocumulus MRS, Cirrocumulus MRS, and Cumulus-white MRS, respectively. That is, about 5% of the breast cancer SNPs were nominally associated at the *p* = 0.05 threshold with the Cirrocumulus MRS, while about 10% of the SNPs were nominally associated with the other MRSs. Two SNPs of the *ESR1* locus were nominally associated with the MRS at the *p* = 0.05 threshold.

## 4. Discussion

We investigated the associations between two types of breast cancer risk scores, namely genetic-based and mammogram-based, and found that they were almost independent with one another. Such independence implies that the PRSs and the MRSs predict breast cancer risk through different pathways so that when they are combined, the two types of risk scores will likely improve the disease risk prediction [31]. This independence of the familial risk scores suggests that the PRS and MRSs additively explain the familial risk of breast cancer. These implications need to be tested empirically.

Twin and family studies suggest that approximately 40–70% of the variance of the MRSs could be explained by additive genetic factors [10,11,12,13]. The findings of this study suggest that the currently known common genetic variants associated with breast cancer risk account for at most a small amount (<1%) of the variance of the MRSs. SNPs found by GWAS of the MRSs only explain about 2–4% of the variance of the Cumulus MRS and Cumulus-percent MRS [19,25].

Taking all the evidence into consideration, the vast majority of the genetic variance of the MRSs remains unexplained. DNA methylation has been proposed to explain part of the genetic variance of human traits, but our previous study did not find evidence for an association of peripheral blood DNA methylation with the Cumulus MRS or Cumulus-white MRS [32]. More studies are needed to discover the genetic and epigenetic determinants of the MRSs.

It is also possible that inherited rare mutations implicating in breast cancer risk, such as *BRCA1* and *BRCA2* germline mutations, could explain part of the variance in the MRSs. However, findings about the association between *BRCA1* and *BRCA2* mutations and Cumulus-percent density measure are inconsistent [33,34]. These studies have small to moderate sample sizes. More studies are needed to investigate this question, especially for the Altocumulus MRS and the Cirrocumulus MRS.

We found that the Altocumulus MRS remained associated with the breast cancer PRSs after adjusting for the Cumulus MRS although the associations decreased by approximately 65–80%. This is consistent with our findings that the association of the Altocumulus MRS with breast cancer risk remained after adjusting for the Cumulus MRS [9]. On the contrary, the associations of the Cumulus MRS with the breast cancer PRS (this study) and breast cancer risk [9] became null after adjusting for the Altocumulus MRS. These findings suggest that not all the association between the Altocumulus MRS and breast cancer polygenetic susceptibility, measured using the PRS, is modulated by the Cumulus MRS.

We found, consistently across the three PRSs, that their associations with the Altocumulus MRS were stronger than with the Cumulus-white MRS. This suggests that the bright, not white, regions of the mammogram are more strongly associated with breast cancer polygenic susceptibility, and studying these regions could reveal more about breast cancer genetic susceptibility. The currently known major predisposition genes and the current PRS discovered to date explain less than half the familial aggregation of breast cancer [35], and new approaches to finding the missing heritability are needed.

As to lifestyle- and hormone-related risk factors, they are weakly correlated in relatives and have a weaker association with breast cancer risk than the MRSs, so they do not explain much familial aggregation of breast cancer [14,29]. Therefore, studying the genetic factors associated with these risk factors is unlikely to find the missing heritability of breast cancer.

From the analysis of the SNPs known to be associated with breast cancer risk, we replicated the previously reported associations with the *ATXN1* and *ESR1* loci [19,21,25]. This analysis also supports an association between breast cancer genetic susceptibility and the MRSs.

## 5. Conclusions

In conclusion, our study suggests that less than 1% of the variance of the MRSs is explained by the genetic markers currently known to be associated with breast cancer risk. Discovering the genetic determinants of the bright, not white, regions of the mammogram could reveal substantial new genetic determinants of breast cancer.

## Figures and Tables

**Table 1 cancers-14-02767-t001:** Characteristics of the study sample combined, and by categories of twins and sisters.

Characteristics and Measurements	Total Sample (*n* = 2559)	MZ Twins (*n* = 1168)	DZ Twins (*n* = 636)	Non-Twin Sisters (*n* = 755)
**Age, BMI, and breast cancer PRS, Mean (standard deviation)**				
Age (years)	54.0 (8.4)	54.2 (8.2)	53.5 (8.9)	54.1 (8.1)
BMI (kg/m^2^)	26.1 (5.2)	25.7 (4.9)	26.4 (5.2)	26.5 (5.7)
PRS for overall breast cancer	−0.42 (0.64)	−0.44 (0.63)	−0.44 (0.64)	−0.36 (0.65)
PRS for ER− breast cancer	−0.32 (0.60)	−0.34 (0.60)	−0.34 (0.62)	−0.28 (0.58)
PRS for ER+ breast cancer	−0.42 (0.68)	−0.43 (0.68)	−0.43 (0.68)	−0.34 (0.70)
**Mammogram density measures, Median (Inter-quartile range)**				
Cumulus (cm^2^)	29.4 (18.2–42.9)	28.4 (17.7–41.1)	29.0 (18.2–42.8)	31.9 (19.0–46.0)
Cumulus–percent (%)	29.6 (16.6–43.2)	29.8 (16.8–42.9)	28.4 (15.7–42.3)	30.0 (17.5–44.2)
Altocumulus (cm^2^)	11.3 (6.7–17.1)	11.1 (6.7–16.1)	11.0 (6.2–17.0)	12.1 (6.9–18.6)
Cirrocumulus (cm^2^)	1.6 (0.7–3.1)	1.6 (0.7–3.0)	1.5 (0.7–3.2)	1.7 (0.7–3.3)
Cumulus-white (cm^2^)	17.3 (10.4–26.5)	16.6 (10.2–25.8)	17.1 (10.7–26.8)	17.9 (10.7–28.0)

**Table 2 cancers-14-02767-t002:** Correlations between mammogram risk scores and breast cancer polygenic risk scores.

Mammogram Risk Scores	PRS for Overall Breast Cancer			PRS for ER− Breast Cancer			PRS for ER+ Breast Cancer		
	Correlation (95% CI)	*p*	BIC	Correlation (95% CI)	*p*	BIC	Correlation (95% CI)	*p*	BIC
Cumulus MRS	0.06 (0.02, 0.10)	2.7 × 10^−3^	7275.68	0.06 (0.02, 0.10)	3.0 × 10^−3^	7275.88	0.05 (0.02, 0.09)	0.01	7277.13
Cumulus- percent MRS	0.06 (0.02, 0.10)	1.8 × 10^−3^	7274.93	0.07 (0.03, 0.11)	2.5 × 10^−4^	7271.22	0.06 (0.02, 0.09)	0.01	7276.64
Altocumulus MRS	0.08 (0.04, 0.12)	5.9 × 10^−5^	7268.50	0.07 (0.03, 0.11)	6.0 × 10^−4^	7272.87	0.08 (0.04, 0.11)	1.1 × 10^−4^	7269.19
Cirrocumulus MRS	0.05 (0.02, 0.09)	0.01	7277.07	0.04 (0.01, 0.08)	0.03	7284.23	0.05 (0.01, 0.09)	0.01	7277.45
Cumulus-white MRS	0.04 (−0.01, 0.08)	0.06	7281.04	0.04 (0.01, 0.08)	0.03	7279.65	0.03 (−0.01, 0.07)	0.10	7282.02

**Table 3 cancers-14-02767-t003:** Associations between mammogram risk scores and breast cancer polygenic risk scores.

Mammogram Risk Scores	PRS for Overall Breast Cancer			PRS for ER− Breast Cancer			PRS for ER+ Breast Cancer		
	Regression Coefficient (95% CI)	*p*	BIC	Regression Coefficient (95% CI)	*p*	BIC	Regression Coefficient (95% CI)	*p*	BIC
Cumulus MRS	0.061 (0.017, 0.105)	6.3 × 10^−3^	7334.37	0.059 (0.015, 0.103)	9.2 × 10^−3^	7335.49	0.056 (0.012, 0.100)	0.01	7335.76
Cumulus- percent MRS	0.060 (0.016, 0.104)	7.4 × 10^−3^	7344.65	0.065 (0.021, 0.109)	3.6 × 10^−3^	7343.22	0.055 (0.011, 0.099)	0.01	7345.95
Altocumulus MRS	0.081 (0.037, 0.125)	2.9 × 10^−4^	7328.35	0.068 (0.024, 0.112)	2.5 × 10^−3^	7334.05	0.078 (0.034, 0.121)	5.1 × 10^−5^	7329.35
Cirrocumulus MRS	0.058 (0.016, 0.101)	7.3 × 10^−3^	7345.52	0.046 (0.003, 0.089)	0.03	7350.05	0.056 (0.014, 0.099)	9.4 × 10^−3^	7345.41
Cumulus-white MRS	0.039 (−0.004, 0.083)	0.08	7342.32	0.045 (0.002, 0.089)	0.04	7341.48	0.033 (−0.010, 0.077)	0.13	7343.28

Covariates in the regression model included the first 10 principal components based on genome-wide SNP data. The regression coefficients were scaled as per standard deviation in PRS adjusted for the covariates.

**Table 4 cancers-14-02767-t004:** Associations between mammogram risk scores and breast cancer polygenic risk scores adjusting for other mammogram risk scores.

Mammogram Risk Scores	Other Mammogram Risk Score(s) Adjusted for	PRS for Overall Breast Cancer		PRS for ER− Breast Cancer		PRS for ER+ Breast Cancer	
		Regression Coefficient (95% CI)	*p*	Regression Coefficient (95% CI)	*p*	Regression Coefficient (95% CI)	*p*
Cumulus MRS	Altocumulus	−0.010 (−0.029, 0.009)	0.30	−0.001 (−0.019. 0.019)	0.99	−0.013 (−0.032, 0.006)	0.17
	Cirrocumulus	0.024 (−0.007, 0.056)	0.13	0.032 (0.0001, 0.063)	0.049	0.021 (−0.011, 0.052)	0.20
	Altocumulus + Cirrocumulus	−0.011 (−0.030, 0.008)	0.25	−0.001 (−0.020, 0.018)	0.89	−0.014 (−0.033, 0.005)	0.14
Cumulus-percent MRS	Altocumulus	−0.021 (−0.067, 0.024)	0.35	0.020 (−0.025, 0.066)	0.38	−0.028 (−0.073, 0.017)	0.22
	Cirrocumulus	0.018 (−0.033, 0.068)	0.50	0.057 (0.006, 0.108)	0.03	0.009 (−0.042, 0.060)	0.73
	Altocumulus + Cirrocumulus	−0.019 (−0.064, 0.025)	0.39	0.023 (−0.022, 0.067)	0.32	−0.026 (−0.071, 0.018)	0.25
Altocumulus MRS	Cumulus MRS	0.026 (0.008, 0.045)	5.8 × 10^−3^	0.015 (−0.004, 0.034)	0.11	0.028 (0.009, 0.047)	3.6 × 10^−3^
	Cirrocumulus MRS	0.035 (0.010, 0.059)	5.4 × 10^−3^	0.033 (0.009, 0.058)	7.6 × 10^−3^	0.033 (0.008, 0.057)	9.0 × 10^−3^
	Cumulus + Cirrocumulus	0.020 (0.006, 0.035)	6.9 × 10^−3^	0.014 (−0.001, 0.028)	0.07	0.021 (0.006, 0.035)	5.3 × 10^−3^
Cirrocumulus MRS	Cumulus	0.015 (−0.014, 0.045)	0.31	0.003 (−0.027, 0.033)	0.85	0.018 (−0.012, 0.047)	0.25
	Altocumulus	−0.009 (−0.032, 0.015)	0.47	−0.012 (−0.036, 0.011)	0.31	−0.007 (−0.031, 0.016)	0.54
	Cumulus + Altocumulus	−0.011 (−0.034, 0.012)	0.36	−0.012 (−0.035, 0.011)	0.30	−0.010 (−0.033, 0.013)	0.39
Cumulus-white MRS	Altocumulus	−0.007 (−0.023, 0.009)	0.42	−0.001 (−0.017, 0.016)	0.96	−0.008 (−0.025, 0.008)	0.31
	Cirrocumulus	0.009 (−0.011, 0.028)	0.40	0.013 (−0.007, 0.033)	0.19	0.006 (−0.014, 0.026)	0.54
	Altocumulus + Cirrocumulus	−0.008 (−0.024, 0.008)	0.33	−0.001 (−0.017, 0.014)	0.86	−0.010 (−0.025, 0.006)	0.24

Other covariates in the regression model included the first 10 principal components based on genome-wide SNP data. The regression coefficients were scaled as per standard deviation increase in PRS adjusted for all the covariates.

## Data Availability

Data are available through a reasonable request to the corresponding author.

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
