# Peer review of "Genetic Aspects of Mammographic Density Measures Associated with Breast Cancer Risk"

_cancers, 2022, doi:10.3390/cancers14112767_

Round 1

Reviewer 1 Report

In the manuscript the authors build on their previous work which defined alto- and cirrocumulus associated measures of mammogram risk scores as more highly predictive of breast cancer risk than Cumulus mammogram risk scores (MRSs). Essentially, understanding the pattern of bright and brightest white regions on mammograms could effectively improve breast cancer risk assessment. Findings that familial correlations of Cumulus MRS in twin pairs was high, suggesting that genetic factors might explain 60-70% of the variance in Cumulus MRS, provided the rationale for the authors to look for associations between MRS and breast cancer associated polygenic risk scores (PRSs) based on 313 single-nucleotide polymorphisms (SNPs). They conclude that PRS and MRS are almost independent of each other and that less than 1% of the variance in MRS is explained by currently known genetic markers of breast cancer risk, meaning that the majority of such genetic variance remains unexplained.

In general, this manuscript is well done and informative for understanding associations between genetic and mammographic density associated breast cancer risk. I have little to add and believe the manuscript is acceptable for publication but have a couple of comments for discussion that might help put these findings in context.

Minor Comments

  1. Given the previous association between genetic factors and the variance in Cumulus MRS and the current findings that PRS and MRS are almost independent, it could very well be due to an incomplete understanding of genetic factors associated with breast cancer risk. However, environmental factors and the accumulation of somatic mutations are also known to contribute significantly to cancer risk. Is it also plausible that variance in MRS may be associated with these additional factors, which may or may not be linked to heritable factors? Or, could the altocumulus regions on mammograms reflect the additive effects of inherited and somatic mutations in driving breast cancer risk associated with breast density? If possible, it might be helpful to discuss the data a little more within this context.

  1. How does the current analysis compare to associations between PRS/MRS with other known risk factors for breast cancer (e.g. alcohol consumption, pregnancy, etc.). In other words, in a broad context, is there already evidence that this type of risk assessment could improve predictions related to mammographic density despite possible unknown genetic determinants? Alternatively, a lack of association between MRS/PRS with other risk factors could strengthen the link between genetic variance and Cumulus MRS.

Reviewer 2 Report

This is a study of mammogram density and breast cancer polygenic risk scores in a population of Australian sisters, many of whom were twins. The subjects did not have breast cancer.  The goal is to see if the risks scores calculated with mammogram appearance correlated with polygenic risk scores and to what extent the  mammogram risk score accounted for heritability in the sample.  Given that there are no breast cancer patients in the study much is based on imputation of the associations of the two risk scores based on  previous analyses

The paper is very difficult to follow.  the objective is not clear to the typical reader and the meaning of the results is opaque.  would it not be simpler just to evaluate the correlation between the two measures in a straightforward way.  Does the PRS score predict mammographic density.  does the MRS predict breast cancer risk.   If not then they act on breast cancer risk through different pathways.      Need to clearly state the objectives and simplify.  Try to put much of the technical detail in the methods and simplify the introduction.  

the inclusion of the Altocumulus MRS and the Cumulus MRs in the same paper leads to confusion. 

The discussion is quite good and summarises the results and interpretation.

Round 2

Reviewer 2 Report

The paper is improved but should be carefully rewritten with regards to clarity and presentation.  the current revision does note go far enough

must rethink the objectives etc.  remove abbreviations.   the first author dies not have the necessary skill set and should enlist all the co-authorr to help with the revision.